

# Integrative species delimitation based on COI, ITS, and morphological evidence illustrates a unique evolutionary history of the genus *Paracercion* (Odonata: Coenagrionidae)

Haiguang Zhang[1], Xin Ning[2,3], Xin Yu[4] and Wen-Jun Bu[2]

[1] College of Life Sciences, Linyi University, Linyi, China
[2] Institute of Entomology, College of Life Sciences, Nankai University, Tianjin, China
[3] Wenlai High School, Shanghai, China
[4] College of Life Sciences, Chongqing Normal University, Chongqing, China

Corresponding author
Xin Yu, lannysummer@163.com

## ABSTRACT

*Paracercion* are common 'blue and black' colored damselflies. We explore the species boundaries of *Paracercion* (Odonata: Coenagrionidae) using ABGD, bPTP, GMYC and Distance-based clustering. We finally got the molecular data of all nine species of *Paracercion*. *P. hieroglyphicum* and *P. melanotum* were combined into one putative species based on cytochrome c oxidase I (COI). However, they were separated into two putative species based on the nuclear segment including ITS1-5.8S-ITS2 (ITS). This suggests the introgression of mtDNA in *Paracercion*. *Paracercion barbatum* and *Paracercion melanotum* can be separated into two species based on COI, whereas they were combined into one putative species based on ITS, which suggests a hybridization event between them. The lower interspecific divergence (COI: 0.49%) between *P. barbatum* and *Paracercion v-nigrum* indicates a recent speciation event in *Paracercion*. *Paracercion sieboldii* and *P. v-nigrum* can be separated into two putative species based on COI, while they were frequently merged into the same putative species based on ITS. This can be explained by incomplete lineage sorting in nDNA. Besides, *P. pendulum* and *P. malayanum* were synonymized as junior synonyms of *P. melanotum*. *P. luzonicum* was confirmed not to belong to *Paracercion*. The possibility of introgression, hybridization, recent speciation and incomplete lineage sorting makes species delimitation, based on molecular data, difficult and complicates understanding of the evolutionary history of *Paracercion*. The discordance in COI and ITS also indicates the value of using markers from different sources in species delimitation studies.

# INTRODUCTION

Correctly identifying species boundaries is important in taxonomic studies. Species delimitation based on molecular data plays an important role in integrative taxonomy (*Satler, Carstens & Hedin, 2013*; *Andujar et al., 2014*; *Kergoat et al., 2015*;

*Huang & Knowles, 2016*; *Noguerales, Cordero & Ortego, 2018*; *Vidigal et al., 2018*; *Zhang et al., 2018*; *Masonick & Weirauch, 2020*; *Zheng et al., 2020*). A variety of species delimitation methods based on molecular data are used in taxonomic and systematic studies for testing species boundaries (*Sites & Marshall, 2003*; *Flot, 2015*). The most widely used methods for species delimitation include ABGD (*Puillandre et al., 2012*), Distance-based clustering (*Meier et al., 2006*; *Brown et al., 2012*), GMYC (*Pons et al., 2006*; *Fujisawa & Barraclough, 2013*), and PTP (*Zhang et al., 2013*). Automatic Barcode Gap Discovery (ABGD) is an automatic procedure that sorts sequences into hypothetical species based on barcode gaps. Distance-based clustering uses pairwise distances for clustering sequences into "molecular operational taxonomic units" (MOTUs) (*Floyd et al., 2002*; *Vogler & Monaghan, 2007*; *Jones, Ghoorah & Blaxter, 2011*). The generalized Mixed Yule Coalescent (GMYC) method is a likelihood method for delimiting species by fitting within and between species branching models to an input tree. The Poisson Tree Processes (PTP) model assumes that the number of substitutions between species is significantly higher than the number of substitutions within species. bPTP is an updated version of PTP that adds Bayesian support values to delimited species on the input tree.

Although multilocus approaches are now common, single-gene methods still predominate in numerous studies (*Dellicour & Flot, 2018*). The COI (cytochrome c oxidase I) gene has been widely used in DNA-based species delimitation studies, especially after the introduction of DNA barcoding (*Hebert et al., 2003*; *Blair & Bryson, 2017*). However, limitations of mtDNA, such as ancestral polymorphism, sex-biased gene flow and incomplete lineage sorting (*Moritz & Cicero, 2004*; *Rubinoff, 2006*), complicate the definition of species boundaries based on mtDNA sequence data. Also, mtDNA introgression has been found in many insect groups (e.g., *Bachtrog et al., 2006*; *Cong et al., 2017*), fishes (*Bernal et al., 2017*), amphibians (*Bryson et al., 2014*; *Kuchta, Brown & Highton, 2018*), reptiles (*McGuire et al., 2007*), birds (*Dai, Dong & Yang, 2020*) and mammals (*Liu et al., 2011*). The discordance between mitochondrial and nuclear genetic markers across taxa is now recognized and erroneous conclusions may occur in species delimitation studies based only on the mitochondrial gene (*Papakostas et al., 2016*). Therefore, multiple loci, including both mitochondrial and nuclear genes, which can provide independent evidence, are better suited for species delimitation. ITS is a nuclear genetic marker for DNA barcoding in botany and fungi, and it is very useful in systematic studies of the Odonata (*Dow, Hämäläinen & Stokvis, 2015*; *Yu et al., 2015*).

*Weekers & Dumont (2004)* established the genus *Paracercion* based on molecular data and morphological characters. *Yu & Bu (2011)* combined *Coenagrion dorothea* Fraser, 1924 as *Paracercion dorothea* (Fraser, 1924) and confirmed that *Paracercion impar* (Needham, 1930), one of the eight species described in *Dumont (2004)*, was a junior synonym of *P. dorothea*. *Ning et al. (2016)* described a new species *Paracercion ambiguum* Kompier & Yu, 2016 from Vietnam based on molecular and morphological characters. Twelve species are recorded in the genus *Paracercion* globally (*Schorr & Paulson, 2020*). They are *P. ambiguum* Kompier & Yu, 2016, *P. barbatum* (Needham, 1930), *P. calamorum* (Ris, 1916), *P. dorothea* (Fraser, 1924), *P. hieroglyphicum* (Brauer, 1865), *P. luzonicum* (*Asahina, 1968*), *P. malayanum* (Selys, 1876), *P. melanotum* (Selys, 1876), *P. pendulum*

(*Needham & Gyger, 1939*), *P. plagiosum* (Needham, 1929), *P. sieboldii* (Selys, 1876), and *P. v-nigrum* (Needham, 1930). Among these, three Southeast Asia species (*P. pendulum*, *P. luzonicum*, and *P. malayanum*) are poorly studied. In the original descriptions, Selys put *P. malayanum* and *P. melanotum* into the same species group 3, and mentioned that the only difference between them was "par les bandes dorsales noires des 1–4 segments decoupees en dessins etroits" in the former but not in the later (*Selys-Longchamps, 1876*). However, after studied hundreds of specimens of *P. melanotum*, the third author XY found both kinds of color patterns occurring in this species. Furthermore, we examined photos of a male specimen of *P. malayanum* in the RMNH (Natural History Museum, London, UK) collection (kindly provided by Dr. Rory A. Dow) and found it identical to male *P. melanotum*. The original description of *P. pendulum* noted that 'this species is nearest *C. barbatum* Needham (*P. barbatum*)......' (*Needham & Gyger, 1939*). However, all the figures (*cf.* Figs. 239,242–243,246, in *Needham & Gyger (1939)*) are virtually identical to *P. melanotum*, even the total blue colored face that is unique to this species. Therefore, we confirmed that both *P. pendulum* and *P. malayanum* are junior synonyms of *P. melanotum*. Furthermore, depending on the original description and figures, especially the shape of male caudal appendages (*cf.* Figs. 51–52 in *Asahina (1968)*), *P. luzonicum* is definitely not a *Paracercion* species but belongs to *Pseudoagrion* or *Coenagrion*. In fact, this species is very similar to *Pseudagrion australasiae*. Finally, with the elimination of *P. pendulum*, *P. malayanum*, and *P. luzonicum*, nine nominal species of *Paracercion* were analyzed in this study.

*Paracercion* are common 'blue and black' colored damselflies distributed in East Asia. They are ubiquitous and common, even in large cities. Previous taxonomic studies were mainly based on morphological characteristics using a limited number of specimens. *Paracercion* species are often convergent in appearance while some of them exhibit large intraspecific variation. Therefore, species delimitation of *Paracercion* based on molecular data with adequate sample sizes was needed. We addressed the taxonomy of this genus by integrating both molecular and morphological data.

## MATERIALS & METHODS

### Taxon sampling

The geographic distribution of *Paracercion* is shown in Fig. 1. A total of 348 individuals of the genus *Paracercion* were collected from China, Japan, and Vietnam. For ITS, 16 sequences of *Paracercion* and seven sequences of out-groups downloaded from NCBI (National Centre for Biotechnology Information, https://www.ncbi.nlm.nih.gov/) were also included in the analyses. In total, nine nominal species of *Paracercion* were analyzed with the molecular data. In addition, the sequences of five genera: *Coenagrion*, *Enallagma*, *Pseudagrion*, *Ischnura* and *Ceriagrion* were chosen as out-groups. The geographic coordinates of different locations were recorded using a global positioning system (GPS) tool. For ease of DNA amplification, most of the specimens were first deposited in anhydrous alcohol in the field and then stored in a freezer at −20 °C at the College of Life Sciences, Chongqing Normal University. The detailed collecting information of the 348 specimens is shown in Table S1.

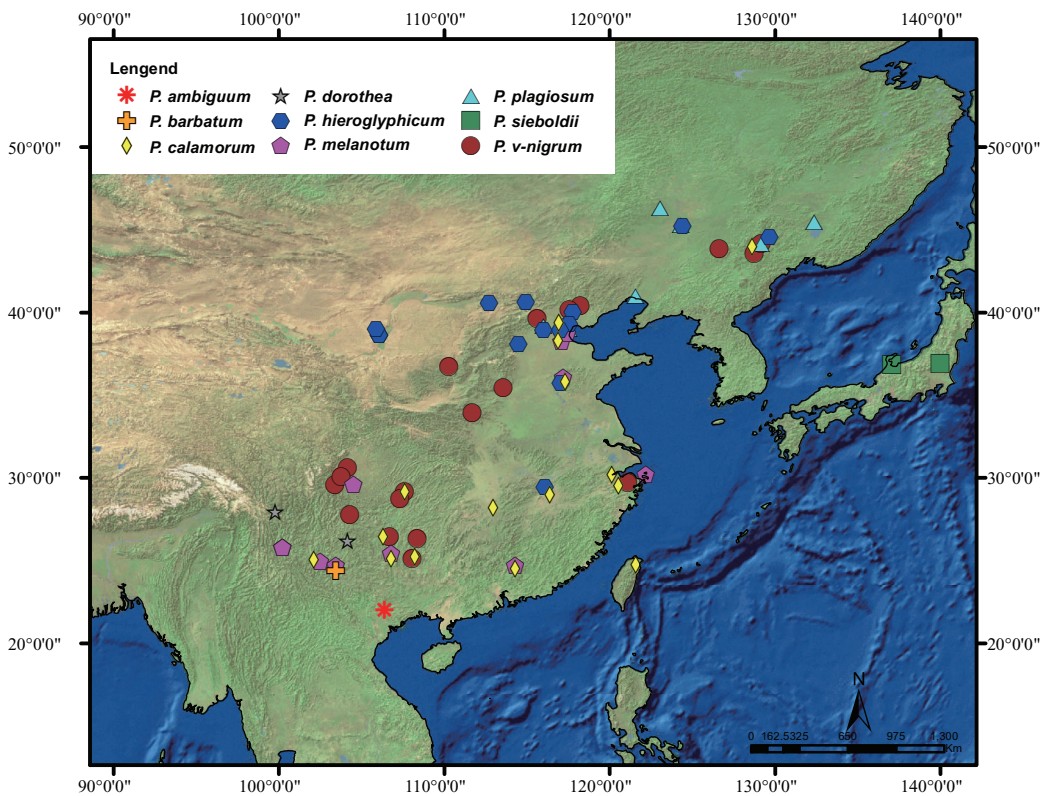

**Figure 1 Distribution of the genus *Paracercion*.** The morphospecies of *Paracercion* are represented by different shapes and colors. Distribution maps were generated using the software ArcGIS 10 (ESRI, Redlands, CA, USA).

## DNA extraction, amplification and sequencing

One hind leg of each specimen was used to extract the genomic DNA by the Universal Genomic DNA Kit (Beijing CoWin Biosciences, Changping, China) following manufacturer protocol. The mitochondrial gene cytochrome c oxidase subunit 1 (COI) and a nuclear segment including ITS1-5.8S-ITS2 (ITS) were first amplified following *Yu et al. (2015)*. However, the amplification success rate of the ITS region was relatively low even using the clone method. All PCR products were visualized using 1% agarose gel electrophoresis and purified using a gel extraction kit (Sangon Biotech, Shanghai, China). The PCR products were sequenced by Beijing Genomics Institute or GENEWIZ based on Sanger's chain termination method.

## Sequence analysis

The sequences files were viewed and edited in BioEdit 7.2.0 (*Hall, 1999*). Alignments of the COI sequences were translated into amino acids using MEGA X (*Kumar et al., 2018*) to detect the presence of pseudogenes. All of the sequences were saved as FASTA format and aligned with MAFFT version 7 (*Katoh & Standley, 2013*).

## OTU delimitations

The ABGD analysis was performed using the web version (http://wwwabi.snv.jussieu.fr/public/abgd/). The detailed arguments were as follows: Pmin = 0.001, Pmax = 0.1, setp = 10, relative gap width = 1.5, Nb bins (for distance distribution) = 20 and the genetic distance model was set to Kimura (K80).

Distance-based clustering analysis was performed with the "tclust" function of the R package spider (*Brown et al., 2012*). Fixed empirical thresholds: 1%, 2%, 2.2% and 3% were selected following *Zhang et al. (2017)*. The genetic distances required for clustering analysis were calculated with the "dist.dna" function of the ape package (*Paradis, Claude & Strimmer, 2004*) in R 3.6.3 (*R Core Team, 2020*) using the K2P model (*Kimura, 1980*).

For GMYC analysis, the ultrametric trees were generated in Beast 2.5.2 (*Bouckaert et al., 2019*). Models of DNA sequence evolution were estimated by MrModeltest v2.3 (*Nylander, 2004*) based on Akaike Information Criterion (AIC). The nucleotide substitution model selected for COI and ITS was GTR+I+G. We used the relaxed lognormal clock model and the coalescent constant population tree prior. For each molecular marker, the length of Markov chain Monte Carlo (MCMC) chains was set to 20,000,000 (ITS) and 50,000,000 (COI) with trees sampled every 1000 generations. The convergence of each run was checked with Tracer v. 1.7 (*Rambaut et al., 2018*). The first 10% of trees of each run were discarded, and the maximum clade credibility tree (the posterior probability limit was set to 0.5) was reconstructed using TreeAnnotator (embed in BEAST package) with the mean node heights option. Afterward, the maximum clade credibility tree was used as input for the GMYC analysis with the splits package (*Ezard, Fujisawa & Barraclough, 2013*) in R. The single-threshold method was used to generate the putative species.

For bPTP analysis, the Maximum-likelihood trees were generated with IQ-TREE Version 1.6.10 (*Nguyen et al., 2015*). We obtained branch supports with the ultrafast bootstrap (*Hoang et al., 2018*) implemented in the IQ-TREE software. ModelFinder (*Kalyaanamoorthy et al., 2017*) was used to compute the log-likelihoods of an initial parsimony tree for different models and to select the best-fitting models of sequence evolution. The out-groups selected for rooting phylogenetic trees were removed in the bPTP analysis. The number of MCMC generations was set to 200,000; the thinning value was set to 100 and the burn in value was set to 0.1. Other parameters were set to default values. The maximum likelihood solution result of bPTP is discussed in this study.

The ultrametric tree and the combined results of different species delimitation methods were visualized by the R package ggtree (*Yu et al., 2017*).

## Network analysis

The sequences files were transformed into haplotype sequences using the R package haplotypes (*Aktas, 2020*) with gaps and missing data included. Thereafter the haplotype network of each molecular marker was reconstructed with PopART 1.7 (*Leigh & Bryant, 2015*) using the TCS method (*Clement, Posada & Crandall, 2000*).

## Morphologic checking

All of the living photos were taken in the field with a digital camera (Nikon D3200) (Bangkok, Thailand). Character photos were taken in the laboratory using the Zeiss V20 (Jena, Germany) microphotography system. Specimens were examined and dissected under a Zeiss V8 stereomicroscope. More than 600 specimens (including dried specimens) were identified by Xin Yu and Xin Ning in terms of the corresponding literatures (*Dumont, 2004*; *Yu, 2008*; *Yu & Bu, 2011*; *Ning et al., 2016*). All the specimens were stored at the College of Life Sciences, Chongqing Normal University. The identification of the *Paracercion* species is not easy due to the prevalent convergent in appearance and intraspecific variations. The most important diagnostic morphologic characters are the shapes of male caudal appendages and genital ligulae and these were carefully examined. Generally, the variation of male genital ligulae among species is limited. This cannot provide more information on species delimitation other than to show some degree of relationship. The structure of the caudal appendages is the most credible diagnostic character. Based on this character, all the individuals can be divided into nine morphological defined species.

# RESULTS

## Molecular data

We obtained molecular data from all the nine species of *Paracercion*. The COI dataset consisted of 348 sequences from *Paracercion and* 25 sequences from out-groups with a length of 1,066 bp. There were 312 variable sites and 282 parsimony-informative sites in 348 COI sequences and most of the variation occurred in the third codon position. The ITS dataset consists of 66 sequences from *Paracercion* and seven sequences from out-groups. We obtained both COI and ITS sequences from all the nine species of *Paracercion*.

## Genetic distance (K2P distance)

The COI genetic distances between *P. hieroglyphicum* and *P. melanotum* ranged from 0–1.65%. In comparison, the ITS genetic distances between *P. hieroglyphicum* and *P. melanotum* ranged from 9.58% to 10.89%. Similarly, the COI genetic distances between *P. barbatum* and *P. v-nigrum* ranged from 0.19% to 1.16%; the ITS genetic distances between *P. barbatum* and *P. v-nigrum* ranged from 3.03% to 8.44%. The COI genetic distances of *P. barbatum* ranged from 0 to 0.19% (average value: 0.08%, Table S2). The ITS genetic distances of *P. barbatum* ranged from 0 to 7.94% (average value: 3.01%, Table S3). Regardless of the specimen, ChH03, the ITS genetic distances of *P. barbatum* ranged from 0 to 3.19%. The COI genetic distances between *P. barbatum* and *P. melanotum* ranged from 9.46% to 10.58%. The ITS genetic distances between ChH03 and 11 specimens of *P. melanotum* ranged from 0.16% to 2.17%; the ITS genetic distances between the other specimens of *P. barbatum* and the 11 specimens of *P. melanotum* ranged from 5.39% to 8.49%. Although the COI genetic distances between *P. sieboldii* and *P. v-nigrum* ranged from 3.13% to 4.05%, the ITS genetic distances between *P. sieboldii* and *P. v-nigrum* were relatively small (0.17–4.13%). The overlap between intraspecific and interspecific genetic

distances indicated that the barcode gap does not exist in the COI and ITS markers
(Fig. S1, Fig S2).

## ABGD analysis

For COI ($P = 0.0129$), 348 specimens were divided into six groups. The specimens of
*P. barbatum*, *P. sieboldii* and *P. v-nigrum* clustered into one group (Group 1).
The specimens of *P. hieroglyphicum* and *P. melanotum* clustered into the same group
(Group 4). The specimens of the other four morphologically defined species: *P. ambiguum*,
*P. calamorum*, *P. dorothea* and *P. plagiosum* were each divided into one of four different
groups (Table S4).

    For ITS ($P = 0.0129$), 66 specimens were divided into seven groups. The specimens of
*P. barbatum* (excluding ChH03), *P. sieboldii* and *P. v-nigrum* clustered into one group.
One specimen (ChH03) of *P. barbatum* and the 11 specimens of *P. melanotum* were placed
in the same group. The specimens of the other five morphologically defined species:
*P. ambiguum*, *P. calamorum*, *P. dorothea*, *P. hieroglyphicum* and *P. plagiosum* were divided
into five different groups (Table S4).

## Distance-based clustering analysis

If the threshold was set to 1%, 2% and 2.2%, the number of MOTUs/clusters defined by
clustering analysis on COI was seven. In comparison, the number of MOTUs for ITS
was 17, 11 and 10 separately (Fig. S3). For COI, if the threshold was set to 3%,
348 specimens could be divided into seven MOTUs. The specimens of *P. barbatum* and
*P. v-nigrum* were grouped into the same MOTU; the specimens of *P. hieroglyphicum* and
*P. melanotum* were also grouped into the same MOTU. The specimens of the other
five morphologically defined species—*P. ambiguum*, *P. calamorum*, *P. dorothea*,
*P. plagiosum* and *P. sieboldii*—were divided into five different MOTUs (Table S5).

    For ITS, if the threshold was set to 3%, 66 specimens could be divided into seven
MOTUs. The specimens of *P. barbatum* (excluding ChH03), *P. sieboldii* and *P. v-nigrum*
were grouped into the same MOTU. One specimen (ChH03) of *P. barbatum* and the
11 specimens of *P. melanotum* were divided into the same MOTU. The specimens of the
other five morphologically defined species—*P. ambiguum*, *P. calamorum*, *P. dorothea*,
*P. hieroglyphicum* and *P. plagiosum*—were divided into five different MOTUs (Table S5).

## GMYC analysis

For COI, 145 haplotypes could be divided into seven putative species (Table 1). The 70
haplotypes of *P. barbatum* and *P. v-nigrum* were grouped into the same putative species;
the 55 haplotypes of *P. hieroglyphicum* and *P. melanotum* were also grouped into the same
putative species. The haplotypes of the other five morphologically defined species:
*P. ambiguum*, *P. calamorum*, *P. dorothea*, *P. plagiosum* and *P. sieboldii* were divided into
five different putative species (Table S6).

    For ITS, 50 haplotypes could be divided into 12 putative species (Table 1).
One haplotype (Hap_4, ChH03) of *P. barbatum* and the nine haplotypes of *P. melanotum*
were included within the same putative species. Six haplotypes of *P. barbatum* were

**Table 1 Species delimitation result of the General Mixed Yule Coalescent (GMYC) method.**

| Marker | LogLnull[a] | LogLGMYC[b] | Clusters[c] | Entities[d] |
|---|---|---|---|---|
| COI | 1236.988 | 1275.555*** | 7 (7–7) | 7 (7–7) |
| ITS | 305.4786 | 308.7191* | 9 (3–14) | 12 (3–24) |

**Notes:**
[a] The log-likelihood of the null model.
[b] Maximum log-likelihood of GMYC model and the result of the likelihood ratio test (***$P < 0.001$, *$P < 0.05$).
[c] Number of ML clusters (excluding singletons).
[d] Number of ML entities (including singletons).

separated into three putative species. Five haplotypes of *P. calamorum* were separated into two putative species. Sixteen haplotypes of *P. sieboldii* and *P. v-nigrum* were separated into two putative species. The haplotypes of the other four morphologically defined species: *P. ambiguum*, *P. dorothea*, *P. hieroglyphicum* and *P. plagiosum* were divided into four different putative species (Table S6).

### bPTP analysis

A total of 145 COI haplotypes could be divided into seven tentative species using bPTP analysis. All the haplotypes of *P. barbatum* and *P. v-nigrum* were grouped into the same tentative species; all the haplotypes of *P. hieroglyphicum* and *P. melanotum* were also grouped into the same tentative species. The haplotypes of the other five morphologically defined species—*P. ambiguum*, *P. calamorum*, *P. dorothea*, *P. plagiosum* and *P. sieboldii*—were divided into five different tentative species (Table S7). The result of bPTP analysis on COI was consistent with the GMYC analysis.

A total of 50 ITS haplotypes could be divided into 10 tentative species by bPTP analysis. One haplotype (Hap_4, ChH03) of *P. barbatum* and the nine haplotypes of *P. melanotum* clustered into the same tentative species. Six haplotypes of *P. barbatum* were separated into three tentative species. Sixteen haplotypes of *P. sieboldii* and *P. v-nigrum* were merged into the same tentative species. The haplotypes of the other five morphologically defined species—*P. ambiguum*, *P calamorum*, *P. dorothea*, *P. hieroglyphicum* and *P. plagiosum*—were divided into five different tentative species (Table S7).

The combined results of species delimitation of different methods based on COI and ITS are shown in Fig. 2 and Fig. 3, respectively.

### Network analysis

The COI haplotype network showed that the haplotypes of *P. barbatum* and *P. v-nigrum* were merged into one large group; however, they did not share any haplotypes. In comparison, the haplotypes of *P. sieboldii* were placed into a single group (Fig. 4A). All the haplotypes of *P. hieroglyphicum* and *P. melanotum* were also grouped into one large group. Two individuals of *P. hieroglyphicum* and two individuals of *P. melanotum* shared the same haplotype (Fig. 4B).

The ITS haplotype network showed that the haplotypes of *P. sieboldii* and *P. v-nigrum* were merged into one nested group; the haplotypes of *P. barbatum* were separated into a
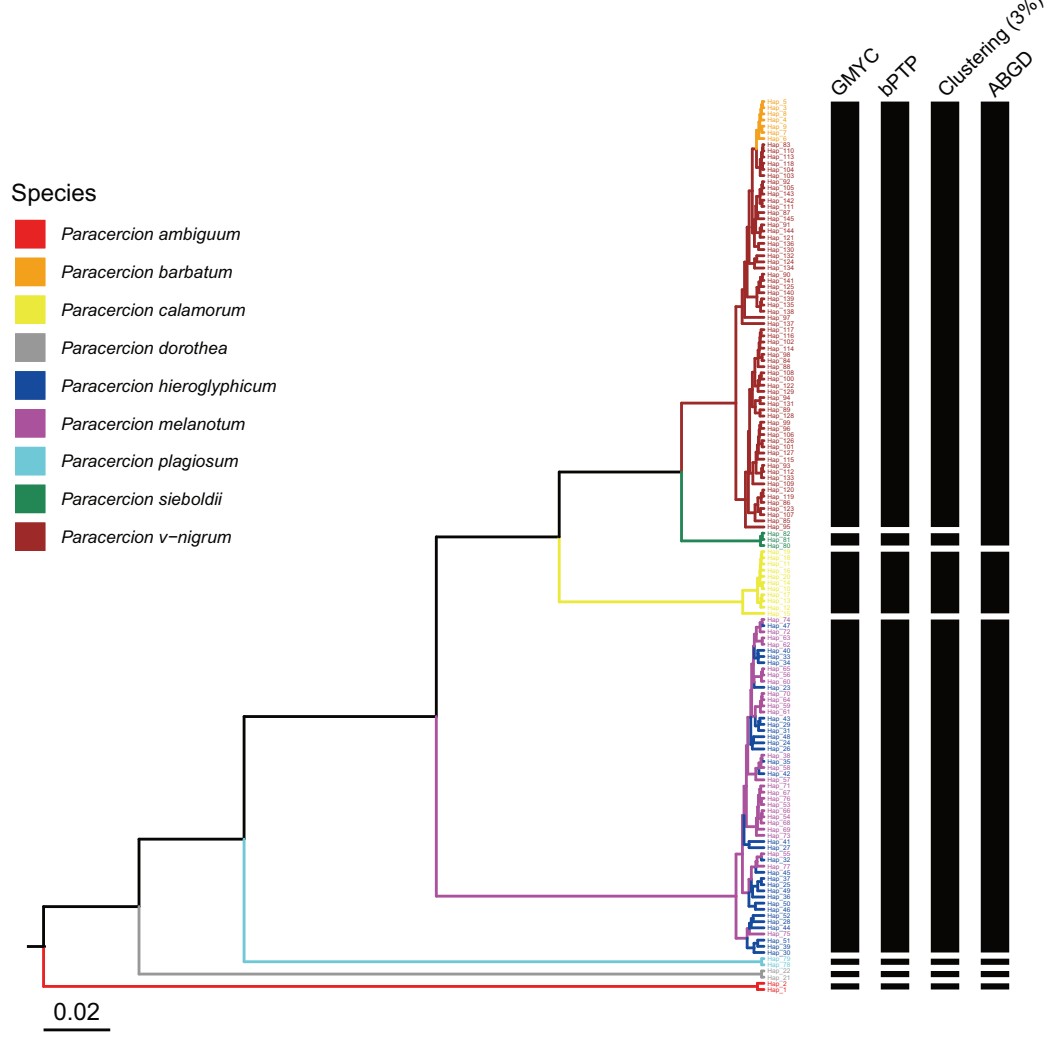

**Figure 2 Comparison of species delimitation results of different methods of *Paracercion* based on the COI haplotypes.** The maximum clade credibility tree generated from a BEAST analysis is colored by nine morphospecies of *Paracercion*.

single group (Fig. 5A). In addition, one haplotype of *P. barbatum* and the nine haplotypes of *P. melanotum* were merged into one group (Fig. 5B).

## DISCUSSION

### Introgression in *Paracercion*

For COI, four methods including ABGD, Clustering, GMYC and bPTP all indicated that *P. hieroglyphicum* and *P. melanotum* belong to one putative species. For ITS, the results of all four methods support the conclusion that these two species are distinct. In addition, there are significant differences between *P. hieroglyphicum* and *P. melanotum* based on morphological characters (Fig. 6), especially the male caudal appendages (Fig. 7). These
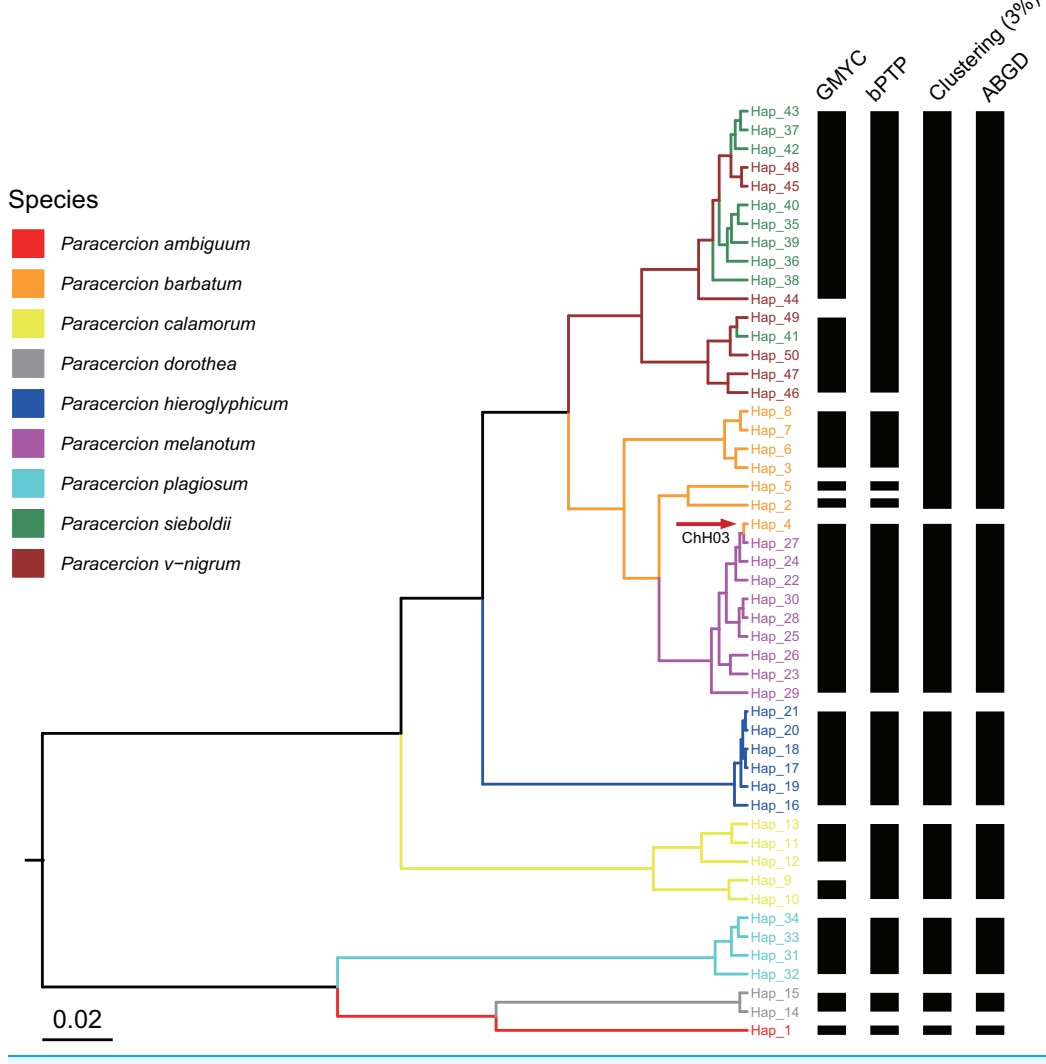

**Figure 3 Comparison of species delimitation results of *Paracercion* using different methods based on the ITS haplotypes.** The maximum clade credibility tree generated from a BEAST analysis is colored by nine morphospecies of *Paracercion*. The hybrid individual (ChH03, Hap_4) was indicated with the red arrow.

differences support the conclusion that these are two separate species. *P. hieroglyphicum* and *P. melanotum* are sympatric species in many areas (Fig. 1) and share similar shaped genital ligulae (Fig. 8). They have identical COI haplotypes (Fig. 4B). The morphological and molecular evidence argue for the introgression of mtDNA in *P. hieroglyphicum* and *P. melanotum*. Our field observations indicate that these two species have evolved reproductive isolation mechanisms. Males sometimes attempt to mate with nonconspecific females, but the latter are not receptive.

In general, introgression in *Paracercion* was found in COI rather than ITS, which indicates that mtDNA may be more likely to show introgression than nDNA (*Ballard & Whitlock, 2004*).

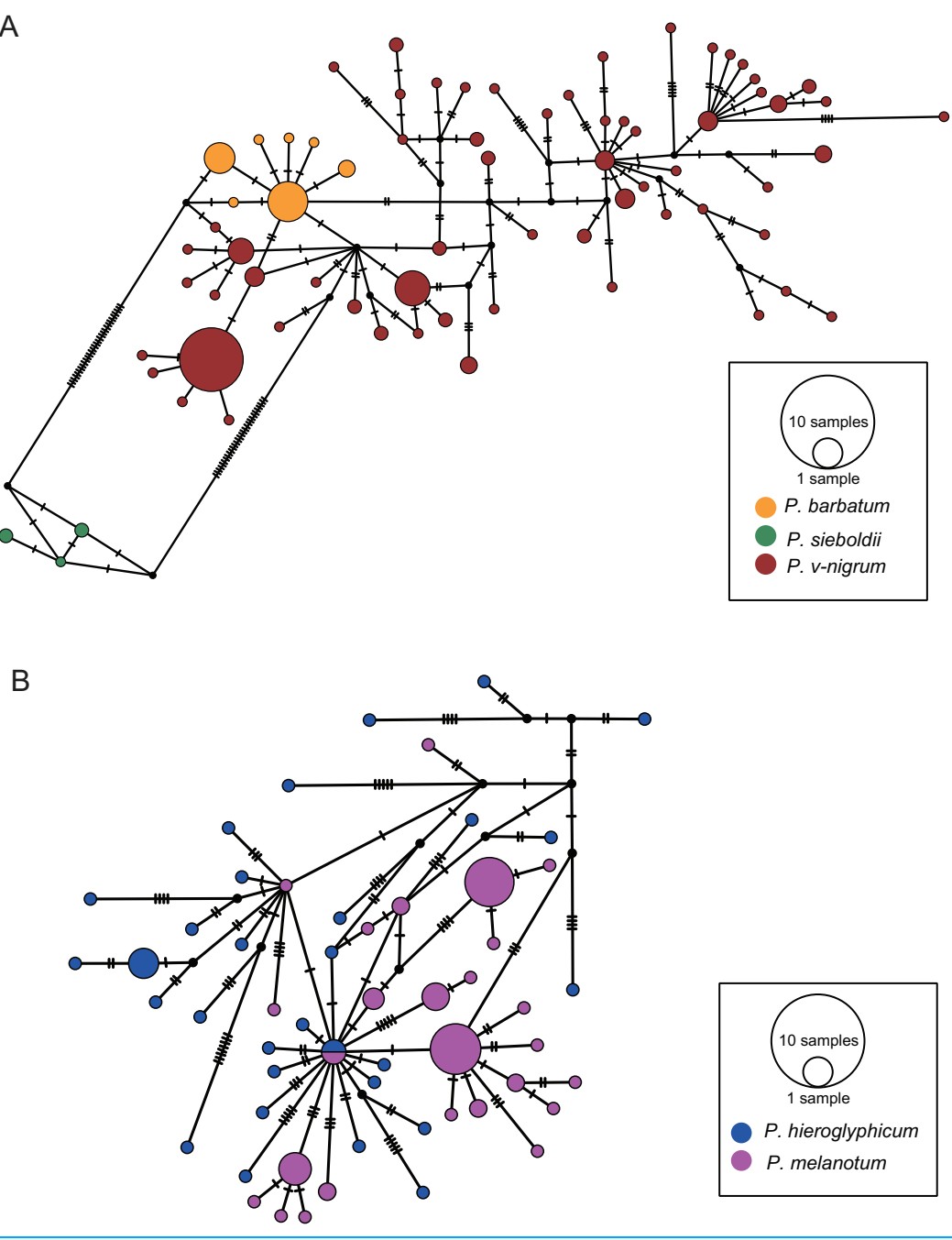

**Figure 4 TCS haplotype networks of different species based on the COI gene.** (A) *P. barbatum*, *P. sieboldii* and *P. v-nigrum*; (B) *P. hieroglyphicum* and *P. melanotum*.

## Hybridization between *P. barbatum* and *P. melanotum*

For ITS, four methods of species delimitation indicated that at least one specimen (ChH03) of *P. barbatum* together with all specimens of *P. melanotum* should be the same tentative species. This indicates that *P. barbatum* and *P. melanotum* likely hybridize at Changhu, Yunnan. Several possible hybrid individuals from Changhu were found based on

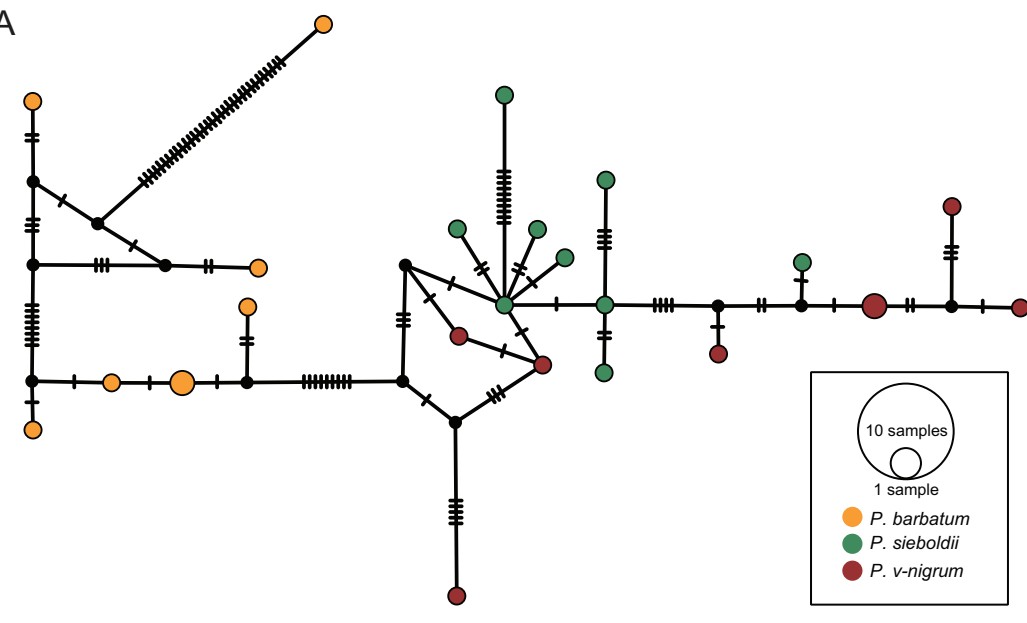

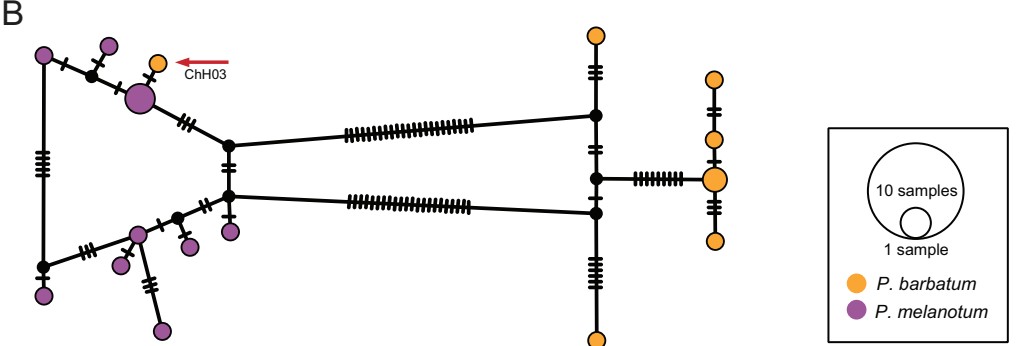

**Figure 5 TCS haplotype networks of different species based on the ITS marker.** (A) *P. barbatum*, *P. sieboldii* and *P. v-nigrum*; (B) *P. barbatum* and *P. melanotum*. The hybrid individual (ChH03) was indicated with the red arrow.

morphological characters (Fig. 9). *P. barbatum* is restricted to Northeast Yunnan while *P. melanotum* is widely distributed (Fig. 1), covering the range of *P. barbatum*. It should be extended further south since the two previous southeast species *P. malayanum* and *P. pendulum* were confirmed to be the same species. Among them, the caudal appendage of a male specimen of *P. malayanum* in the RMNH (Natural History Museum, London, UK) collection (kindly provided by Dr. Rory A. Dow) (Fig. 7O) is identical to male *P. melanotum*. Shapes of the male caudal appendages and genital ligula of *P. barbatum* are similar to that in *P. melanotum* (Fig. 9) and that may be why Needham and Gyger confused them. This also suggests the possibility of hybridization. It is worth noting that recent hybridization events can be detected by ITS rather than COI.
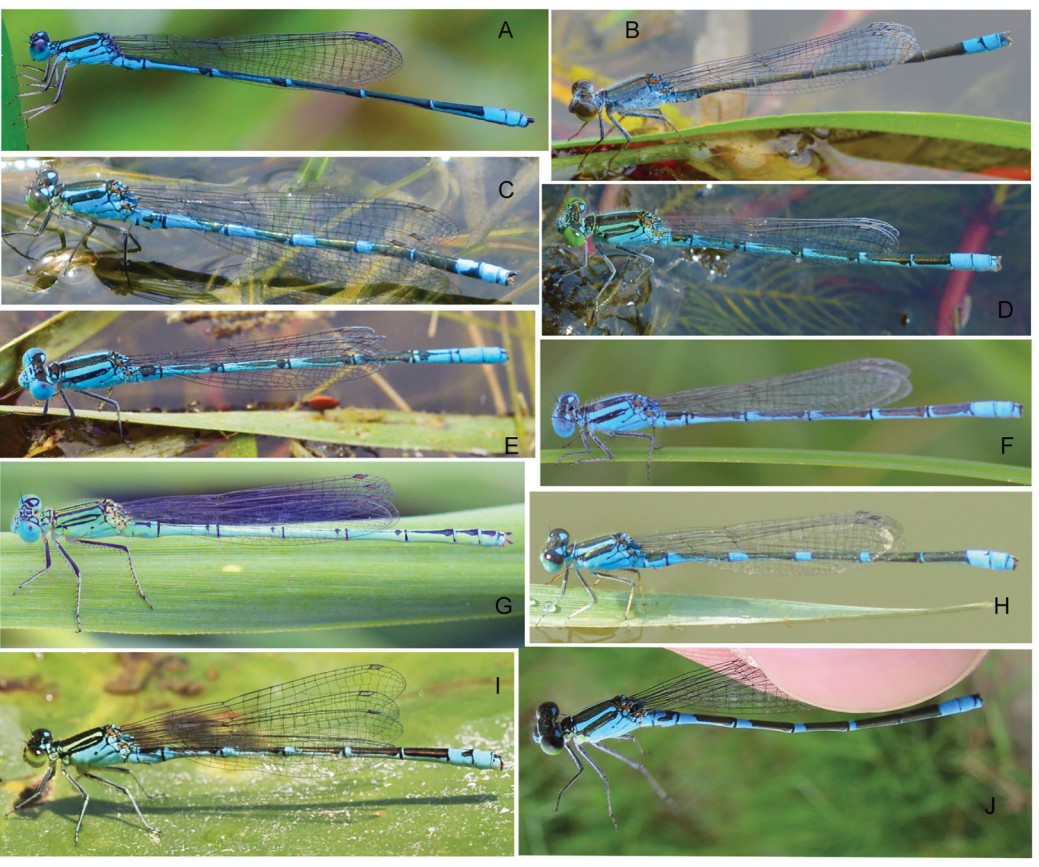

**Figure 6 Habitus of *Paracercion* spp.** (A) *P. ambiguum*; (B) *P. calamorum*; (C) *P. dorothea*; (D) *P. hieroglyphicum*; (E–F) *P. melanotum*, show color pattern varitation; (G) *P. plagiosum*; (H–I) *P. v-nigrum*, show color pattern varitation; (J) *P. barbatum*.

## Species group of '*v-nigrum*'

Results from both COI and ITS supported a very closely related species group including *P. barbatum*, *P. sieboldii* and *P. v-nigrum*. For COI, four methods—ABGD, Clustering, GMYC and bPTP—all indicated that *P. barbatum* and *P. v-nigrum* are a single putative species. The low mean interspecific divergence between *P. barbatum* and *P. v-nigrum* (COI: 0.49%) indicates a recent speciation event in *Paracercion*. For ITS, the results of GMYC and bPTP support that this pair of species were separated into two or more putative species. Furthermore, *P. barbatum* seems to be closely related to *P. melanotum* due to the hybridization discussed above. All the specimens of *P. sieboldii* and *P. v-nigrum* can be separated into two putative species in the four different methods based on COI. However, they were merged into the same putative species in ABGD, Clustering and bPTP analyses based on ITS. This discordance of species delimitation may be the result of incomplete lineage sorting in the nuclear DNA.

*P. barbatum*, *P. sieboldii* and *P. v-nigrum* are allopatric. However, their distributions are obviously contiguous, namely, *P. v-nigrum* has the widest distribution range while

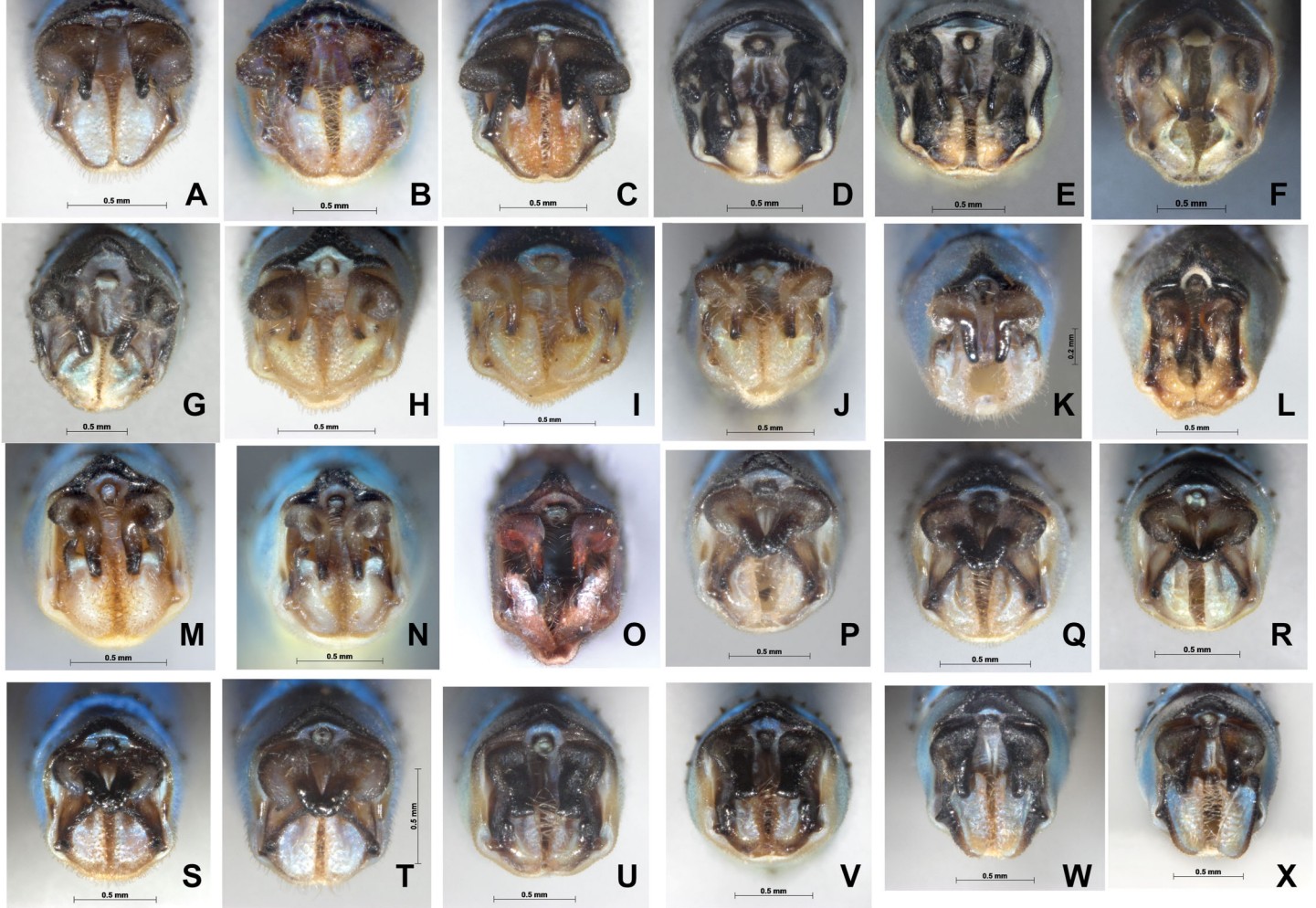

**Figure 7** **Male caudal appendages (rear view) of *Paracercion* spp. from different populations.** Male caudal appendages (rear view) of *Paracercion* spp. from different populations. (A–C) *P. calamorum*: (A) Shandong, (B) Heilongjiang, (C) Yunnan; (D–E) *P. dorothea*, Yunnan; (F–G) *P. plagiosum*: (F) Tianjin, (G) Heilongjiang; (H–J) *P. hieroglyphicum*: (H) Shandong, (I) Tianjin, (J) Ningxia; (K–N) *P. melanotum*: (K) Shandong, (I) Tianjin, (M–N) Yunnan; (O) *P. malayanum* specimen of RMNH (Photo credit: Rory A. Dow); (P–T) *P. v-nigrum*: (P) Beijing, (Q) Sichuan, (R) Heilongjiang, (S) Guizhou, (T) Guangxi; (U–V) *P. barbatum*, Yunnan; (W–X) *P. sieboldii*, Japan.

*P. barbatum* is restricted to the southwest corner (northwest Yunnan) and *P. sieboldii* is isolated at the northeast portion (Japan) of this area (Fig. 1). *P. barbatum*, *P. sieboldii* and *P. v-nigrum* share similar morphological characters especially the shape of genital ligulae (Fig. 6). Therefore, these three species may have originated from one widely distributed ancestral species. *Dumont (2004)* and *Ning et al. (2016)* also suggested that *P. sieboldii* shared a common ancestor with the continental *P. v-nigrum*.

## Species delimitation methods

All methods of species delimitation will sometimes fail to delimit species boundaries properly or will give conflicting results. The use of qualitative judgment will then be

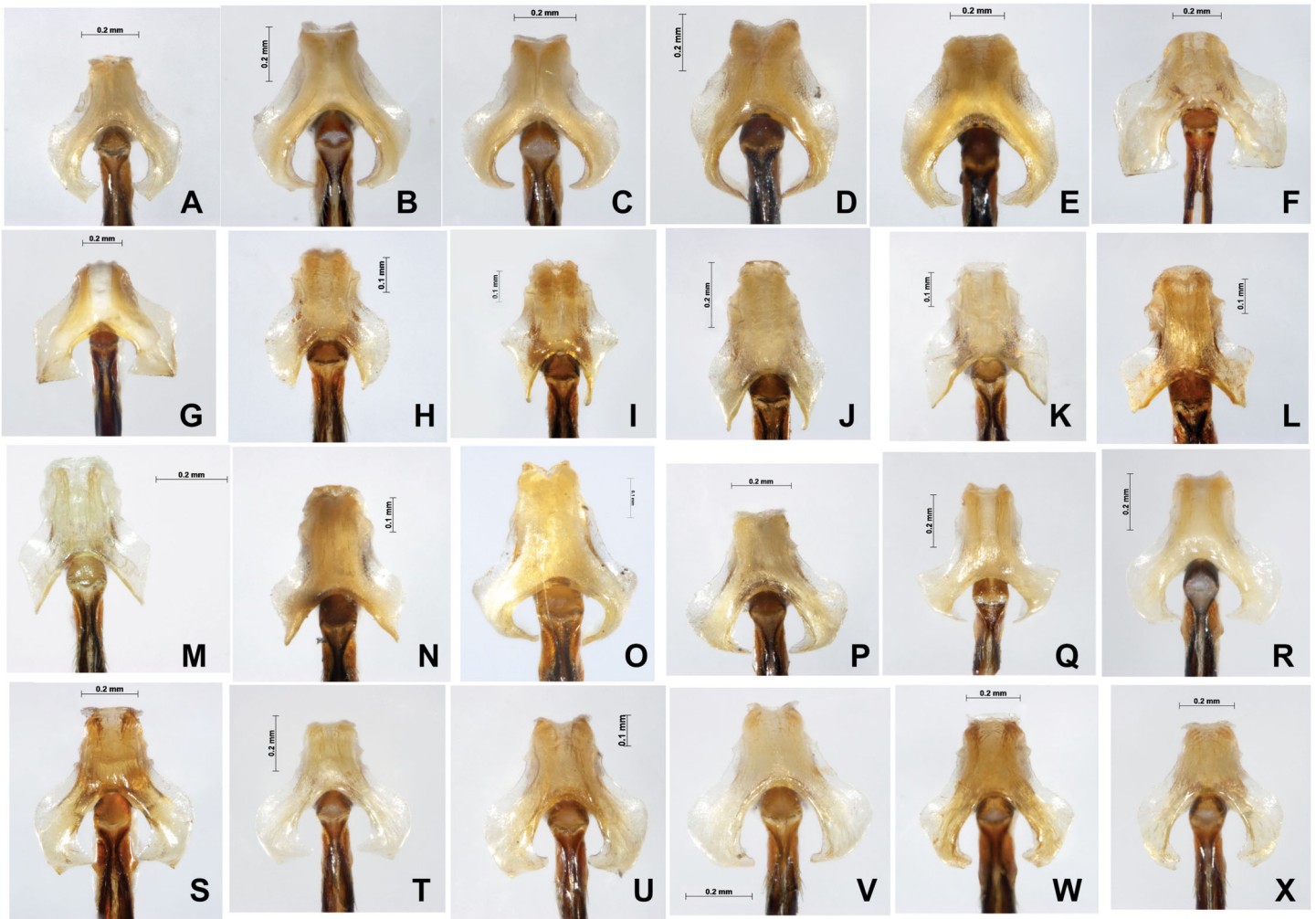

**Figure 8 Male genital ligulae (ventral view) of *Paracercion* spp. from different populations.** (A–C) *P. calamorum*: (A) Shandong, (B) Heilongjiang, (C) Yunnan; (D–E) *P. dorothea*, Yunnan; (F–G) *P. plagiosum*: (F) Tianjin, (G) Heilongjiang; (H–J) *P. hieroglyphicum*: (H) Shandong, (I) Tianjin, (J) Ningxia; (K–N) *P. melanotum*: (K) Shandong, (L) Tianjin, (M–N) Yunnan; (O) *P. ambiguum*, Vietnam; (P–T) *P. v-nigrum*: (P) Beijing, (Q) Sichuan, (R) Heilongjiang, (S) Guizhou, (T) Guangxi; (U–V) *P. barbatum*, Yunnan; (W–X) *P. sieboldii*, Japan.

necessary (*Sites & Marshall, 2004*). For species delimitation studies, multiple methods based on different principles and multiple loci underlying different evolutionary history should be included. Theoretically, the most realistic strategy is to accept delimitations that are congruent across methods and may represent actual evolutionary lineages. However, morphological characters are also important in species delimitation studies. Inferences drawn from species delimitation studies should be conservative (*Carstens et al., 2013*). For ITS, the number of putative species generated from bPTP and GMYC exceeded the total number of species determined from morphological characteristics. This was mainly due to the high rates of false positives of PTP and GMYC methods (*Luo et al., 2018*).

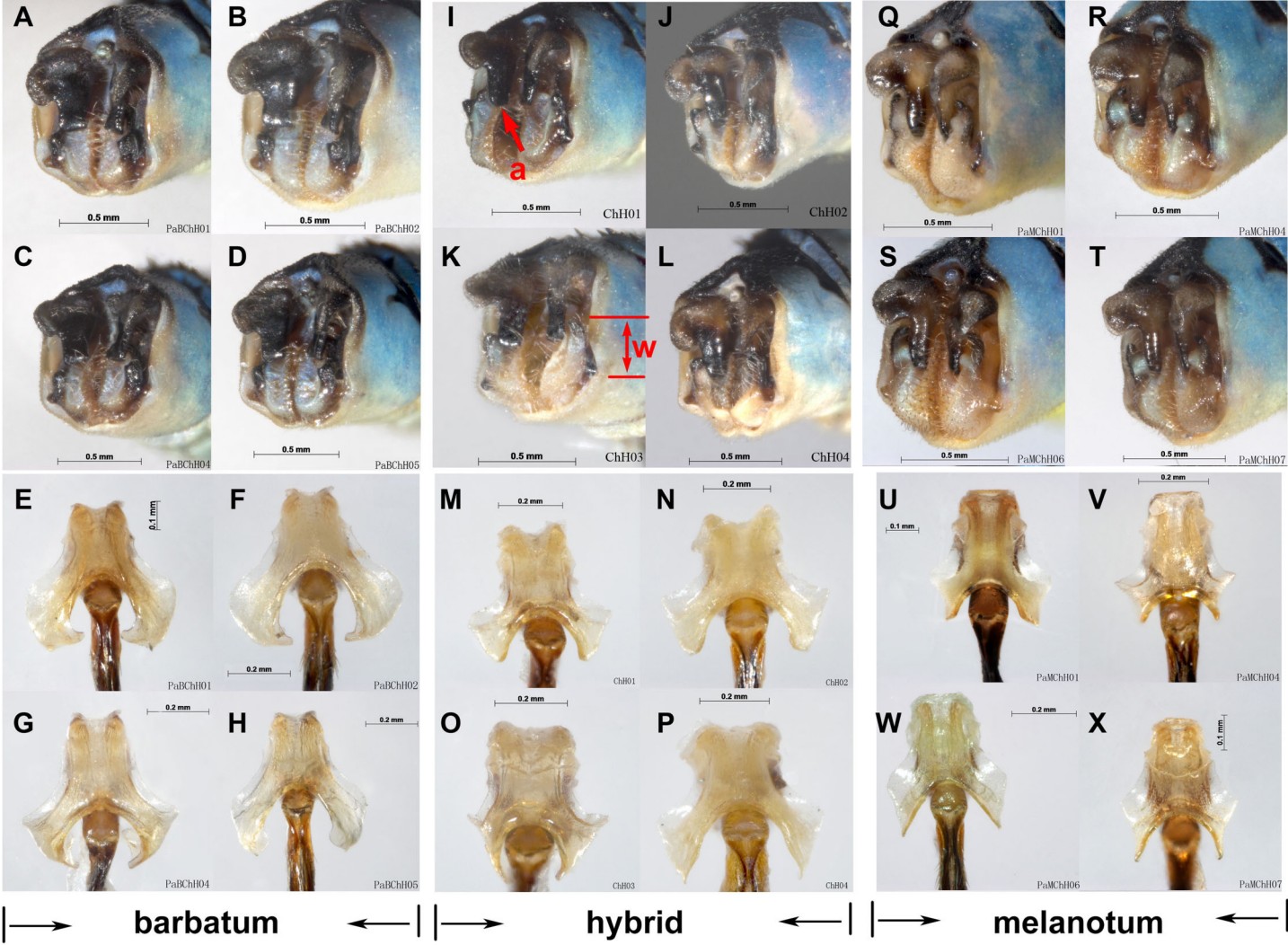

**Figure 9 Hybridization between *P. barbatum* and *P. melanotum* in Changhu, Yunnan.** (A–H) Male caudal appendages and genital ligulae of *P. barbatum*; (I–P) Male caudal appendages and genital ligulae of hybrid individuals; (Q–X) Male caudal appendages and genital ligulae of *P. melanotum*. The most obvious characters showing the intermediate state of the hybrid are the shape of the end of the basal tooth on the cercus (a) and the distance between the two lateral teeth on the paraproct (w). The general shape of genital ligulae also supports hybridization. All the individuals came from Changhu, the hybridization area.

## CONCLUSIONS

Six or Seven tentative species were indicated based on COI with four different methods, while seven, ten, or twelve tentative species were indicated based on ITS. Herein, neither COI nor ITS supported the nine species based on the morphological evidence.

In *Paracercion*, species delimitation results based on ITS were relatively more reliable than those based on COI. *P. pendulum* and *P. malayanum* were synonymized as junior synonyms of *P. melanotum*. *P. luzonicum* was confirmed not to belong to *Paracercion*. Combing both the morphological and molecular evidence, we insist on the genus *Paracercion* includes nine confirmed species.

The possibility of introgression in COI causes difficulties for species delimitation based on molecular data. This phenomenon, together with the discovery of the possible hybridization and incomplete lineage sorting illustrates the unique evolutionary history of the genus *Paracercion*. Anyway, to confirm the existence of above introgression, hybridization, recent speciation, and incomplete lineage sorting, further analysis may need to conduct in the future.

The discordance of the delimitations based on the COI and ITS markers demonstrates the importance of using both mitochondrial and nuclear genes in species delimitations studies. However, species delimitation based on one or two DNA markers may be questionable. Therefore, if possible, more genes should be used in species delimitation studies. Morphological characters and molecular data are complementary. If the results based on morphological characters and molecular data can match each other, the conclusions will be more reliable. Meanwhile, molecular taxonomy may also lead to a false conclusion without support from morphological characters.

## ACKNOWLEDGEMENTS

We are grateful to Prof. Fumio Hayashi and Mr. Hong-guang Jin for providing important specimens. We also thank to Dr. Rory A. Dow for helping us with valuable literatures and photo of Southeast Asia species.

### Funding

This project was supported by the Chongqing Basic Research and Frontier Exploration Special Project (No. cstc2018jcyjAX0415) and the grant of Ministry of Science and Technology of China (No. 2015FY210300). The funders had no role in study design, data collection and analysis, decision to publish, or preparation of the manuscript.

### Grant Disclosures

The following grant information was disclosed by the authors:
Chongqing Basic Research and Frontier Exploration Special Project: cstc2018jcyjAX0415.
Ministry of Science and Technology of China: 2015FY210300.

### Competing Interests

The authors declare that they have no competing interests.

### Author Contributions

- Haiguang Zhang analyzed the data, prepared figures and/or tables, authored or reviewed drafts of the paper, and approved the final draft.
- Xin Ning performed the experiments, analyzed the data, authored or reviewed drafts of the paper, and approved the final draft.
- Xin Yu conceived and designed the experiments, prepared figures and/or tables, authored or reviewed drafts of the paper, and approved the final draft.
- Wen-Jun Bu authored or reviewed drafts of the paper, and approved the final draft.

## DNA Deposition

The following information was supplied regarding the deposition of DNA sequences:

The sequences are available at GenBank: MW361892–MW361941 and MW361519–MW361891.

## Data Availability

The haplotype file of the COI and ITS are available in the Supplemental File.

## Supplemental Information

Supplemental information for this article can be found online at http://dx.doi.org/10.7717/peerj.11459#supplemental-information.

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
