# Peer review of "Integrative species delimitation based on COI, ITS, and morphological evidence illustrates a unique evolutionary history of the genus Paracercion (Odonata: Coenagrionidae)"

_PeerJ, doi:10.7717/peerj.11459_

## Round 0.1 · original submission · Major Revisions

While the topic of the manuscript is interesting, both reviewers have major questions about transparency of the Materials & Methods, the logics of some of your discussion and conclusions, and the layout of your figures. I have the similar feeling and agree with their comments. The authors should carefully revise the manuscript based on their comments and my extra comments below, especially pay attention to improving the clearness and transparency of your methods by adding details, and avoiding self-contradictory conclusions or discussion without support from your results. When resubmit your files, a point-by-point response letter is needed.

1) Fixed four empirical thresholds were used (Line137-138) following Zhang et al. (2017), which was about a hemipteran group. Do the authors think using same thresholds is OK for one mt gene and a nuclear marker? Do the authors have related evidence to show these thresholds are appropriate for your study group? Or, can previous studies on Coenagrionidae or Odonata species suggest any empirical thresholds?

2) Both 'geographic' and 'geographical' are used in the manuscript, the authors should use only one of them. And I suggest the authors also check other '-c' & '-cal' words, if there are.

Reviewer 1 ·

Basic reporting

1. The color labels in Figure 2 and Figure 3 for presenting different species may make readers puzzled. The colors for different species and for different analysis methods have some overlaps. Besides, it is hard to distinguish the colors for P. hieroglyphicum and P. sieboldii.

2. It was declared that "species delimitation based on a single gene may be questionable", however, species delimitation based on two genes/markers may still be questionable.

3. Only one mitochondrial marker and one nuclear marker were used in this work, but the conclusion was made as "The discordance in mtDNA and nDNA data ...".

4. For the conclusion "The discordance in mtDNA and nDNA data also indicates the value of using multilocus datasets in species delimitation studies", can two markers be viewed as multiloci?

5. The photo of a key species, P. barbatum was absent in Figure 8.

6. For the hybrid individuals shown in Figure 9, they should be indicated correspondingly in Figure 2, Figure 3, Figure 4, and Figure 5.

Experimental design

From Figure 1, it seems that the sampling for P. melanotum and P. v-nigrum were not enough according to the geological distribution. To show the robustness of sampling, the analyses of ecological niche reconstruction should be made.

Validity of the findings

1. It was concluded that "Our results indicated that the genus Paracercion includes nine confirmed species." But in lines 82-83, it was demonstrated that "We now think that P. pendulum and P. malayanum should both be junior synonyms of P. melanotum". Viewpoint can not be used as evidence.

2. All of the so called "introgression, hybridization, recent speciation and incomplete lineage sorting" are speculations. They should not be taken for sure as "The existence".

Additional comments

1. The study should focus on the relationships between P. v-nigrum, P. barbatum, and P. melanotum rather than on the whole genus, as some species were not included.

2. It is not that necessary to make too many speculations of introgression, hybridization, recent speciation, and incomplete lineage sorting and thus make the study appear meaningful in broad sense. In strict sense, each of the above speculations needs more data and analyses to serve as evidence.

Reviewer 2 ·

Basic reporting

The abstract needs more specific details. How many species were analyzed (out of how many in the genus)? What, specifically, were the “multiple methods” used for species delimitation, both in terms of data and in terms of analysis?

The sentence on line 83 needs support: either provide evidence or a citation. The sentence on line 87 does not follow from what came before. If the authors “now think” think that P. pendulum and P. malayanum are junior synonyms of P. melanotum, why not actually establish that fact in the present manuscript?

***The material and methods section needs to specify how the specimens were identified, who identified them, what resources were used to identify them, and the collection depository of the specimens. Ideally, the specimen data would be digitized and made available on-line. I am not an expert on Odonata, but it might be useful to describe how easy or hard it is to identify the specimens to species. Without this information, how is the reader to know that the results weren’t just based on incorrect specimen determinations?

The sentence on line 103 should specify that these 16 and 8 sequences were in addition to those acquired in the lab for this present study.

***GenBank accession numbers need to be included for the DNA sequences, probably in Table S1.

The authors report on the number of variable and informative sites for CO1, but do not say anything about ITS (line 184). Also, were ITS sequences acquired for all nine species? Line 182 does not specify if it refers to one or both loci.

Split the sentence beginning line 223 into two. The first sentence reporting on CO1 (include the following sentence describing the 3% threshold, the second sentence reporting on ITS.

The “Morphological checking” paragraph (lines 285-290) should probably be included in the introduction and/or material and methods. Then, the molecular methods can be set up as a test of the morphological concepts.

In the figures, the colors for P. hieroglyphicum and P. sieboldii need to contrast better: I can’t tell them apart.

Experimental design

The experimental design is appropriate.

Validity of the findings

The findings are valid.

Additional comments

The manuscript is a valuable contribution to the field. The research is presented clearly and succinctly and highlights the challenges of molecular species delimitation.

The conclusions are somewhat self-contradictory, however. The authors state that we should recognize species based on a congruence of methods (line 356), but they favor splitting the putative taxa over lumping them: that is, they recognize species delimited by CO1 data that are not delimited by ITS data, and they recognize species delimited with ITS that are not delimited with CO1. If they suggest that congruence is important, shouldn’t they only recognize species that are delimited with both datasets? In the end, in recognizing nine species (line 365), they validate the congruence of the morphological data with either, but not necessarily both, of the molecular datasets. Personally, I think this approach is appropriate, but the authors should clarify their discussion accordingly (especially the “Species delimitation methods” section and the “Conclusions”).

---

## Round 0.2 · Minor Revisions

Although the manuscript has been improved in some parts, the two reviewers still have their own concerns about your writing logic, explanation of results, and conclusions. Especially Reviewer 1 insists on some points as previously mentioned, which you should seriously deal with in a new revision.

Reviewer 1 ·

Basic reporting

no comment

Experimental design

no comment

Validity of the findings

no comment

Additional comments

The authors have made a clear thing seem complex and puzzling.
In the conclusion, they declared that "our results indicated that the genus Paracercion includes nine confirmed species". However, 6 or 7 bars were indicated by the results based on COI in Figure 2, while 7, 10, or 12 bars were indicated by the results based on ITS in Figure 3. That means neither COI nor ITS well supported the conclusion of 9 species, which was in fact based on morphological evidence. Certainly, compared with the results based COI, those based on ITS are more close to the value 9, as this value was covered by the range from 7 to 12. Therefore, it is necessary for the authors to really make major revisions to their whole manuscript from the title to the conclusion part. As for the title, "Species delimitation based on a single gene may be questionable" is nothing new and too story-telling. It is better to have a more serious title such as "Improved species delimitation based on COI, ITS, and morphological evidence: ...".

Reviewer 2 ·

Basic reporting

Abstract:“For ITS, Paracercion barbatum and Paracercion melanotum can be combined into one putative species, which suggests a hybridization event between them.” It seems this sentence is missing information. Is it the fact that CO1 did separate the two nominal species whereas ITS did not that suggests a hybridization event?

There is some confusion as to the reporting of the two new synonyms and new combination.
The introduction should present the fact that all 12 nominal species were examined but that P. luzonicum was eliminated after preliminary analysis. Therefore, 11 nominal species were analyzed with the molecular data. If they want, the authors can mention the fact that two of those species are understudied and questionable, and even state their validity as among the hypotheses to be tested in the present study.
The materials and methods should then reassert the fact that 11 nominal species were examined.
The results, in turn, state that P. pendulum and P. malayanum were genetically indistinguishable from P. melanotum. Specimens putatively identified as these nominal species should be indicated in the various figures so the authors can see how they are clearly embedded within P. melanotum.
Finally, the discussion should propose the formal synonymy of these three nominal species (two junior synonyms and one senior).
The abstract should include mention of the removal of P. luzonicum from the genus and the establishment of the two new synonymies.

The species color codes should match across all figures. For example, P. barbatum is purple in Fig 1, orange in Figs 2 and 3, and red in Figs 4 and 5: make them all the same color.

Experimental design

The experimental design is appropriate.

Validity of the findings

The findings are valid and the discussion improved from the previous version.

Additional comments

Although the text is understandable, the English could use some editing and the syntax and message could be improved to make the article more readable and much more cited. “Decimation” (line 317) means something completely different than “delimitation”.

---

## Round 0.3 · accepted · Accept

I think the authors have addressed the questions from the reviewers in this revision. Now the manuscript can be accepted for publication. This paper will provide a useful example of integrative taxonomy.